# Structure of full-length cobalamin-dependent methionine synthase and cofactor loading captured *in crystallo*

Johnny Mendoza [1,6], Meredith Purchal[2,3,6], Kazuhiro Yamada[1,4] ✉ & Markos Koutmos [1,2,5] ✉

Cobalamin-dependent methionine synthase (MS) is a key enzyme in methionine and folate one-carbon metabolism. MS is a large multi-domain protein capable of binding and activating three substrates: homocysteine, folate, and *S*-adenosylmethionine for methylation. Achieving three chemically distinct methylations necessitates significant domain rearrangements to facilitate substrate access to the cobalamin cofactor at the right time. The distinct conformations required for each reaction have eluded structural characterization as its inherently dynamic nature renders structural studies difficult. Here, we use a thermophilic MS homolog (*t*MS) as a functional MS model. Its exceptional stability enabled characterization of MS in the absence of cobalamin, marking the only studies of a cobalamin-binding protein in its apoenzyme state. More importantly, we report the high-resolution full-length MS structure, ending a multi-decade quest. We also capture cobalamin loading *in crystallo*, providing structural insights into holoenzyme formation. Our work paves the way for unraveling how MS orchestrates large-scale domain rearrangements crucial for achieving challenging chemistries.

B[12]-cofactors and their cobalamin (Cbl) derivatives catalyze a range of challenging biological processes, such as methyl-transfer[1–5], carbon-skeleton rearrangement[3,6,7], dehalogenation[8], essential for complex life, including the biosynthesis of amino acids and $CO_2$ conversion[1,9]. The ability to mediate these diverse chemistries derives from the reactivity of the central cobalt ion (Co)[10,11] (Fig. 1a) and enzymes dependent on this prosthetic group have developed strategies to control its unique organometallic properties[2,12]. Cobalamin-dependent methionine synthase (MS) is an essential enzyme that leverages the reactivity of Cbl to catalyze three distinct methyl-transfer reactions (Fig. 1b)[1]. MS is central to one-carbon metabolism, serving as the enzymatic link between the folate and methionine metabolic cycles[1,13,14], and defects in MS result in severe pathologies including megaloblastic anemia, substantial birth abnormalities, such as neural tube defects[15–17].

MS catalyzes the formation of H[4]folate (tetrahydrofolate, THF) and methionine (MET) from CH[3]-H[4]folate (methyltetrahydrofolate, MTF) and homocysteine (HCY), respectively (Fig. 1b, e, Supplementary Fig. 1). To achieve these challenging reactions, MS uses one of the strongest nucleophiles found in nature, Co(I)[18,19], to abstract a methyl group from a tertiary amine in MTF, a weak methyl donor, to yield H[4]-folate (THF) and methylcob(III)alamin [MeCbl, CH[3]-Co(III)] (Reaction II, Fig. 1b, e)[18,20,21]. The methyl group is transferred from MeCbl to HCY, an unreactive thiol at neutral pH, to form methionine and regenerate the Co(I) state of the cofactor (Reaction I, Fig. 1b, e)[22–24]. Thus, MS reactivity relies on the enzyme continuously cycling between the CH[3]-Co(III) and Co(I) forms of the cofactor[1,25]. Once every ~2000 turnovers microaerophilic cellular conditions lead to the inactivation of MS, where the Co(I) supernucleophile is oxidized to a catalytically inactive

[1]Department of Chemistry, University of Michigan, Ann Arbor, MI 48109, USA. [2]Program in Chemical Biology, University of Michigan, Ann Arbor, MI 48109, USA. [3]New England Biolabs, Inc., Ipswich, MA 01938, England. [4]Department of Biological Chemistry, University of Michigan, Ann Arbor, MI 48109, USA. [5]Program in Biophysics, University of Michigan, Ann Arbor, MI 48109, USA. [6]These authors contributed equally: Johnny Mendoza, Meredith Purchal. ✉e-mail: yamadak@umich.edu; mkoutmos@umich.edu

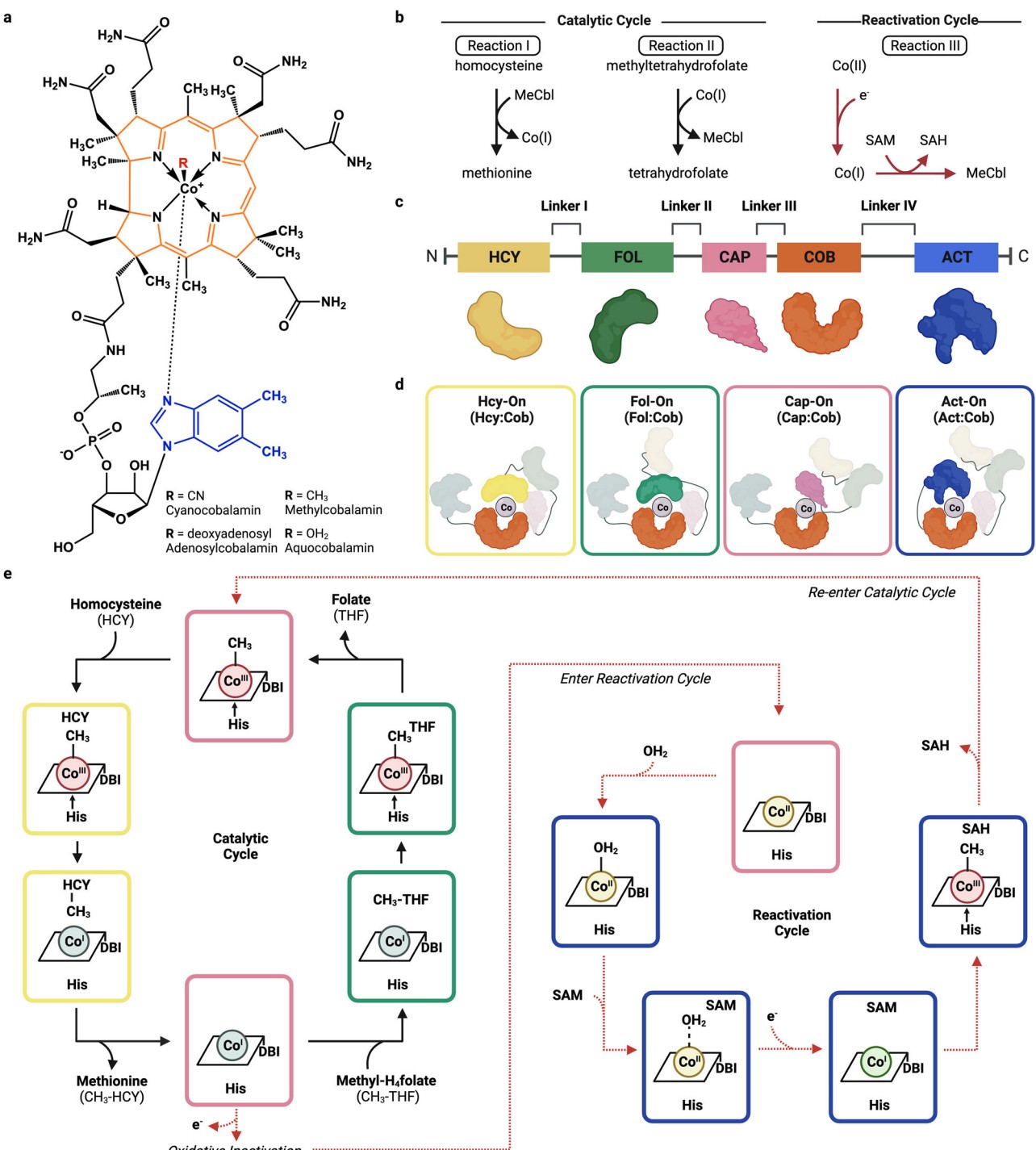

**Fig. 1 | Modular organization and dynamic conformations of methionine synthase (MS).** **a** The structure of the cobalamin cofactor and its various biologically important/relevant forms. **b** The three methylation reactions performed by methionine synthase. **c** Domain organization of modules in MS, including the homocysteine binding domain (Hcy, yellow), folate binding domain (Fol, green), Cap (pink), cobalamin binding domain (Cob, red-orange), and the activation domain (Act, blue). Cartoon representations of each domain are shown above. **d** Predicted MS conformations that must be adopted for catalysis and reactivation. **e** The catalytic cycle of MS. The cobalamin cofactor (simplified to the main scaffold) is methylated by MTF to generate the catalytically active MeCbl, which methylates HCY to MET. The Cbl(II) state is inactive; SAM and an electron source regenerate MeCbl in a scavenge pathway. Panels **c** and **d** were created with BioRender.com.

Co(II) species[13,26]. To restore catalytic competency, MS undergoes a reactivation cycle where Co(II) is enzymatically reduced and subsequently methylated by *S*-adenosyl-ʟ-methionine (AdoMet, SAM) to regenerate the functional MeCbl cofactor that can re-enter the catalytic cycle (Reaction III, Fig. 1b, e)[26–30]. In addition to cycling between different oxidation states, the coordination environment of the cofactor is tightly controlled by association and dissociation of

different axial ligands to achieve each reaction, such as the interplay between the nucleotide dimethylbenzimidazole (DBI) tail and a coordinating His residue on the lower axial position (α face) and the varying upper axial ligand including either a methyl group, water, and no ligand (Supplementary Fig. 2a, b)[31–34].

To support these chemically distinct methyl transfer reactions, MS is synthesized as a single polypeptide chain with long, flexible

linkers connecting 5 domains−akin to beads on a string (Fig. 1c)[35,36], with specialized domains that bind each requisite substrate (HCY, MTF, and SAM) and the Cbl cofactor (Fig. 1d)[24,26,35–38]. The conserved cobalamin-binding domain (Cob, red-orange) carries Cbl, and the adjacent Cap domain (pink) protects the reactive cofactor from unwanted side reactions as the protein cycles through its catalytic steps (Fig. 1d, Cap-On)[39]. The homocysteine-binding domain (Hcy, yellow) activates HCY for methylation and the closely associated folate-binding domain (Fol, green) fosters the methyl transfer from MTF to Co(I), forming THF and completing the catalytic cycle. The reactivation cycle is supported by the activation domain (Act, blue), which binds SAM to regenerate active MeCbl via reductive methylation after its inactivation to the Co(II) form (Fig. 1d, e)[26,40]. As such, each step of the catalytic and reactivation cycles requires a different domain to access the upper face of the B$_{12}$ cofactor; this is achieved by a series of coordinated large-scale rearrangements to orient each substrate-binding domain above the B$_{12}$ cofactor binding domain−deemed 'molecular juggling' (Fig. 1d)[33,41,42]. Thus, MS must adopt at a minimum four unique conformations; the B$_{12}$ domain will cycle between two ternary catalytic conformations (Fol-On and Hcy-On in Fig. 1d) that support the two methyl transferase reactions in the primary catalytic cycle (Reactions I and II), a transition or resting state by adopting the Cap-On conformation, and the Act-On confirmation to support the third methyltransferase reaction between B$_{12}$ and SAM (Reaction III). However, the catalytically relevant conformations required for each reaction (Fig. 1d) have eluded structural characterization.

Given the multi-modular nature of this enzyme, understanding the dynamic molecular motions required to load, protect, and activate cobalamin to catalyze each reaction is a significant challenge. The use of excised domains in a 'divide-and-conquer' strategy has been thus far required to structurally characterize MS due to significant challenges in obtaining and crystallizing the entire protein[22,37,39–46]. During the writing of this manuscript, a tetradomain structure of MS with Cbl bound was captured using Cryo-EM and SAXS (8G3H, Hcy:Fol:Cap:Cob) in a so-called "resting state" (Cap-on), with the Cap:Cob domain nudged in between the Hcy:Fol domains[47]. While this represents the first tetradomain structure of MS, the low resolution of the structure and the inherent flexibility of MS raised further questions regarding cofactor loading, along with the different domain arrangements required to accommodate the three distinct methylation reactions in the full-length enzyme. While the work by Watkins et al.[47]. certainly represents a significant contribution, further work is required to allow for insights into the nature of the apoenzyme state but also to allow us to interrogate how full-length MS binds its cofactor and the role it plays in guiding structural rearrangements.

The knowledge gap regarding structural and mechanistic insights into MS function is largely due to substantial biochemical challenges in working with MS from traditional sources, including homologs from *Homo sapiens* (*H. sapiens*), *Escherichia coli* (*E. coli*), and *Thermotoga maritima* (*T. maritima*). Indeed, the first structure of an MS domain (Cap:Cob, 1BMT) was published in 1994[38], representing the first structure of Cbl bound to a protein, yet, almost three decades later, no full-length structure of MS exists. While progress on how the substrates themselves are loaded has been studied structurally using excised domains, such as the N-terminal Hcy:Fol half (3BOL, 4CCZ) and the C-terminal half Cap:Cob:Act domains (1K7Y, 3BUL), along with the Fol domain (5VON) and Act domain (2O2K, 1MSK, 6BM5)[36,43,48], the loading of the Cbl cofactor has been impossible due to the inability to purify MS in its apo-form. To alleviate these challenges, we sought out alternative sources of MS that would allow us to capture the catalytic conformations of MS, as well as the full-length structure, to facilitate robust characterization of MS dynamics and catalysis.

To that end, we selected cobalamin-dependent MS from *Thermus thermophilus* (*t*MS) as a model system, as thermophiles have long been appreciated for their enhanced stability which can facilitate the structural characterization of challenging proteins[49–54]. Using this model, we successfully captured the full-length, high-resolution structure of MS in its apo-form and captured Cbl cofactor loading in crystallo. The original apo-full-length structure reveals the global orientation of each domain, resembling the reactivation conformation (Act-On) and displaying the flexibility associated with the linker regions that connect each module. Overall, the apo-full-length structure shows that each substrate and cofactor binding site is ready to bind its respective ligand, requiring minimal structural rearrangements. However, nuanced local rearrangements of interest are observed, including around the catalytic and the cobalamin-ligating residue, His761, which can tune the reactivity of cobalamin[40,55]; His761 is found in unique orientation upon Cbl incorporation as captured in crystallo. These findings expand our knowledge of the subtly complex and dynamic role of this catalytic residue and its potential to connect changes in the Cbl cofactor with concomitant conformational rearrangements. The ability to purify and crystallize *t*MS in its apo-form has finally allowed for structural interrogation of Cbl binding, and in crystallo loading provides the final missing piece into how this multi-modular enzyme binds each respective ligand. This has opened the door for exploring fundamental research into how MS and other multi-modular proteins use domain motions and controlled structural remodeling, in the form of local and global structural changes, to exert substrate control and chemical outcome, and the mechanistic properties that allow MS to catalyze three fundamentally distinct methylations.

## Results

### A unique thermophilic MS homolog for studying *apo* metalloenzyme activity and cobalamin loading

The *Thermus thermophilus* HB8 genome encodes for a five-module cobalamin-dependent MS that shares the same domain architecture as *H. sapiens* and *E. coli* MS (Supplementary Fig. 3). Despite relatively high levels of sequence conservation between these homologs (33% and 34% sequence identity, respectively), the systematic challenges typically associated with the purification of MS are entirely absent in *T. thermophilus* MS (*t*MS). Remarkably, we can robustly express and purify the apoenzyme (Fig. 2a) and exogenous Cbl can be incorporated in a simple post-purification heat step to obtain an active enzyme. Furthermore, *t*MS is highly amenable to protein engineering and biochemical manipulation, with all generated mutants being successfully purified. Additionally, each of the *t*MS modules can be expressed and purified individually and as combined penta-, tetra-, tri-, and di-modules (Fig. 2b).

Intrigued by the apparent versatility of this system, we sought to determine if the catalytic properties of full-length *t*MS are comparable to that of human MS (*h*MS) or *E. coli* MS (*e*MS). We first evaluated the thermostability of apo-*t*MS, which remained in the soluble fraction up to 60 °C (Fig. 2a). Upon the addition of MeCbl, *t*MS remained soluble at 70 °C, highlighting the stabilizing effect of cofactor incorporation (Fig. 2a).

In the MeCbl homocysteine methyltransferase reaction (Reaction I, Fig. 1b), the product is cob(I)alamin which is susceptible to inactivation, forming cob(II)alamin when oxidized and/or aquacob(III)alamin in solution[56]. Thus, the performance of the enzyme can be monitored through UV−Vis spectroscopy, taking advantage of the characteristic absorbance features associated with CH$_3$-Co(III) (MeCbl, 520 nm) and its dealkylation to form cob(I)alamin that is rapidly converted to cob(II)alamin (480 nm) under aerobic conditions. By tracking the absorbance change of *holo*-*t*MS upon the addition of substrate HCY to form MET, a clear decrease at 520 nm corresponding to consumption of MeCbl and a coinciding increase at 480 nm associated with cob(II)alamin formation was observed (Fig. 2c), indicating that MeCbl was demethylated, shifting from the Co(III) state to a Co(II) one.

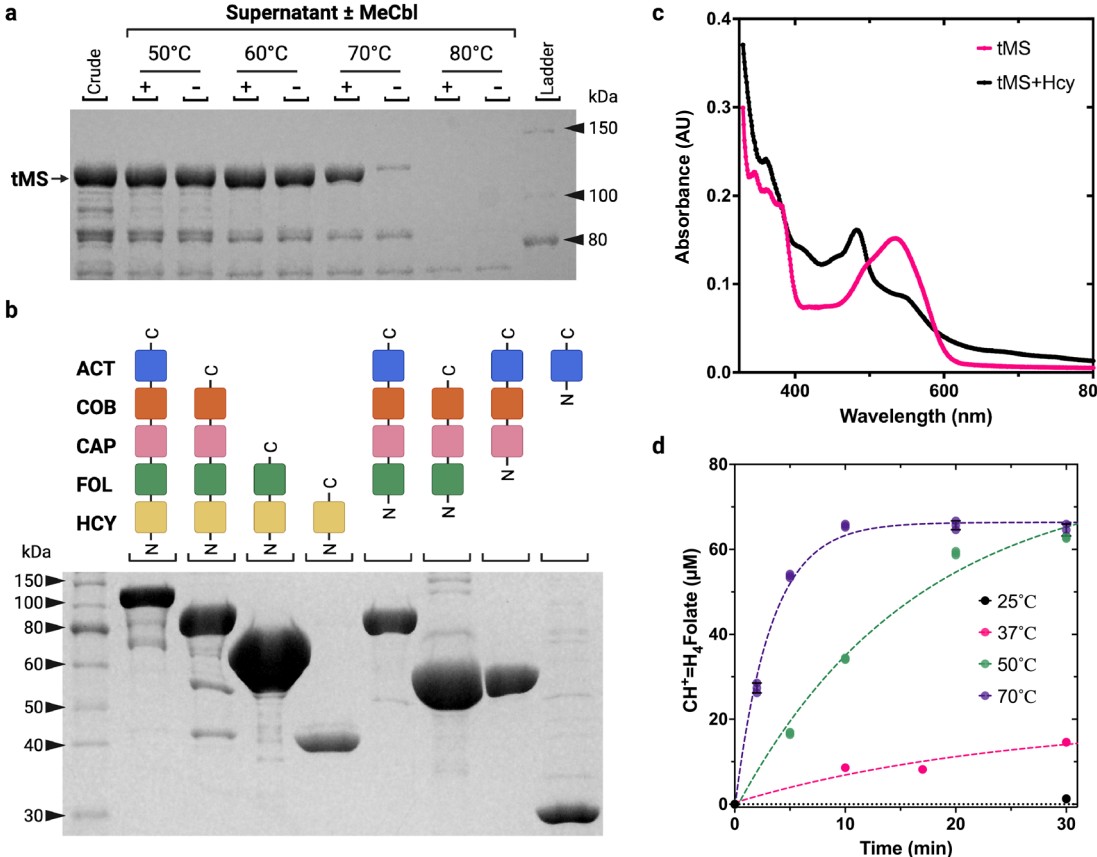

**Fig. 2 | Thermostability and initial biochemical characterization of *t*MS.**
**a** Thermostability of *E. coli* crude extract containing overexpressed full-length *t*MS. Extracts were incubated for 15 min in the presence or absence of 100 μM MeCbl and subject to SDS−PAGE analysis. The gel is a representative experiment, which has been repeated ≥3 times. **b** SDS−PAGE analysis of purified *t*MS excised domains. All excised domain constructs were expressed and purified, highlighting the robustness of *t*MS as a biochemical model. The gel is a representative experiment, which has been repeated ≥3 times. **c** Spectra of methylated and de-methylated of MeCbl bound to full-length *t*MS. *Holo-t*MS was prepared using MeCbl and full-length *apo-t*MS (pink line). The holoenzyme was incubated with excess homocysteine at 25 °C in the presence of dithiothreitol (black line). **d** Effect of temperature on catalytic turnover. A fixed amount of *t*MS was used, and the reaction mixture was incubated for up to 30 min. The product, $H_4$-folate, was converted to methenyltetrahydrofolate ($CH^+=H_4$folate), which was measured at each time point by tracking the absorbance at 350 nm. Data was measured using $n = 2$ biologically independent samples. Source data are provided as a Source Data file.

In the methyltetrahydrofolate methyltransferase reaction (Reaction II, Fig. 1b, e), cob(I)alamin forms $CH_3$-Co(III) by abstracting a methyl group from methyltetrahydrofolate (MTF), forming THF as a product. *t*MS activity can be quantitatively assessed using a coupled steady-state assay, derivatizing the THF byproduct from MTF demethylation to generate methenyltetrahydrofolate ($CH^+=H_4$folate) in situ, which can be tracked by UV–Vis through its characteristic absorbance at 350 nm (Supplementary Fig. 4). To probe the effect of temperature on *t*MS activity, *t*MS was incubated at various temperatures with the requisite substrates for multiple turnover and reactivation (HCY, MTF, and SAM). *t*MS was moderately active at 37 and 50 °C, and most active at 70 °C, as expected for a thermophilic protein (Fig. 2d). To probe substrate dependence on enzymatic turnover, HCY and MTF concentrations were varied respectively and independently; the coupled assay revealed $K_M$ values for MTF and HCY of $18 \pm 4.1$ and $9.3 \pm 3.1$ μM, respectively (Fig. 3a, b and Supplementary Table 1).

In the reactivation cycle and specifically the cob(II)alamin SAM methyltransferase reaction (Reaction III, Fig. 1b, e), SAM is an absolute requirement for reactivation of the catalytically inactive Co(II) via reductive methylation. Thus, the presence of SAM in in vitro MS assays affects the overall observed rate, rescuing MS activity after oxidative inactivation[57]. To probe the SAM dependence of *t*MS activity, namely reactivation after inactivation, the concentration of SAM was varied, and the initial rates were plotted as a function of SAM concentration. The coupled assay revealed a $K_M^{app}$ of approximately $1.0 \pm 0.1$ μM

(Fig. 3c, d and Supplementary Table 1). Overall, *t*MS activity and affinity are comparable to human and *E. coli* MS (Supplementary Table 1) [$k_{cat}$ (min$^{-1}$) of 1062 at 50 °C for *t*MS versus 1542 for *e*MS and 354 *h*MS at 37 °C]. These results suggest that *t*MS is an excellent functional model for methionine synthase, with the ease of its purification as compared to *h*MS making it highly amenable to biochemical studies, and displays similar functional properties to previously characterized MS homologs. In addition, to our knowledge, *t*MS is the only example of an MS construct that can be purified and obtained in its *apo*-form, after which the holoenzyme can be reconstituted at will. We reasoned that we could leverage the versatility of *t*MS, specifically the remarkable stability of *t*MS in the absence of cofactor (apoenzyme) and under high temperatures (up to 80 °C), to structurally interrogate cofactor loading, with our aim to characterize highly sought after but previously unobtainable structures.

## High-resolution insights into the global domain organization of a full-length methionine synthase
To that end, we crystallized full-length 132 kDa *t*MS in its *apo*-form and solved the structure to 2.78 Å resolution (Fig. 4, Supplementary Table 2). *t*MS adopts a compact, wreath-like form, and each domain aligns well with previously determined structures of equivalent excised domain or tridomain constructs (Supplementary Fig. 6). The active site of the helmet-shaped Act domain is positioned above the Cob domain, where the cofactor would sit, similar to the position of these domains

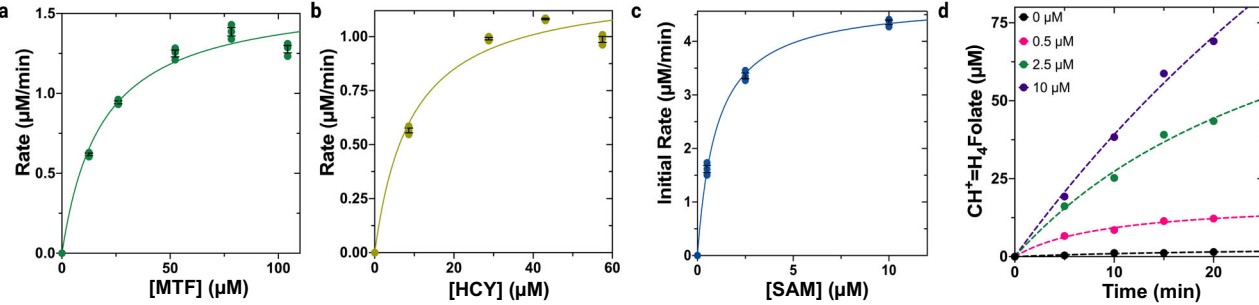

**Fig. 3 | Kinetic and biochemical characterization of *t*MS activity. a** Activity rates based on a CH₃-H₄folate concentration-dependent steady-state assay. A concentration of 100 μM homocysteine was used. The reaction was carried out at 50 °C. **b** Activity rates based on a homocysteine concentration-dependent in the steady-state assay. A concentration of 250 μM CH₃-H₄folate was used. The reaction was carried out at 50 °C. **c** SAM-dependence of *t*MS activity. Product formation under varying concentrations of SAM (0–10 μM). The initial rates were plotted against the

initial concentration of SAM. **d** Time-dependent product formation as a function of SAM concentration (0–10 μM). Product formation was measured for up to 20 min. Data in panels **a**–**c** were measured using *n* = 3 biologically independent samples for the experiments and analyzed using the mean ± SEM. Data in panel **d** is of a representative experiment, which has been repeated ≥3 times, using *n* = 3 biologically independent samples, except for 0 μM, which was done using *n* = 2 biologically independent samples. Source data are provided as a Source Data file.

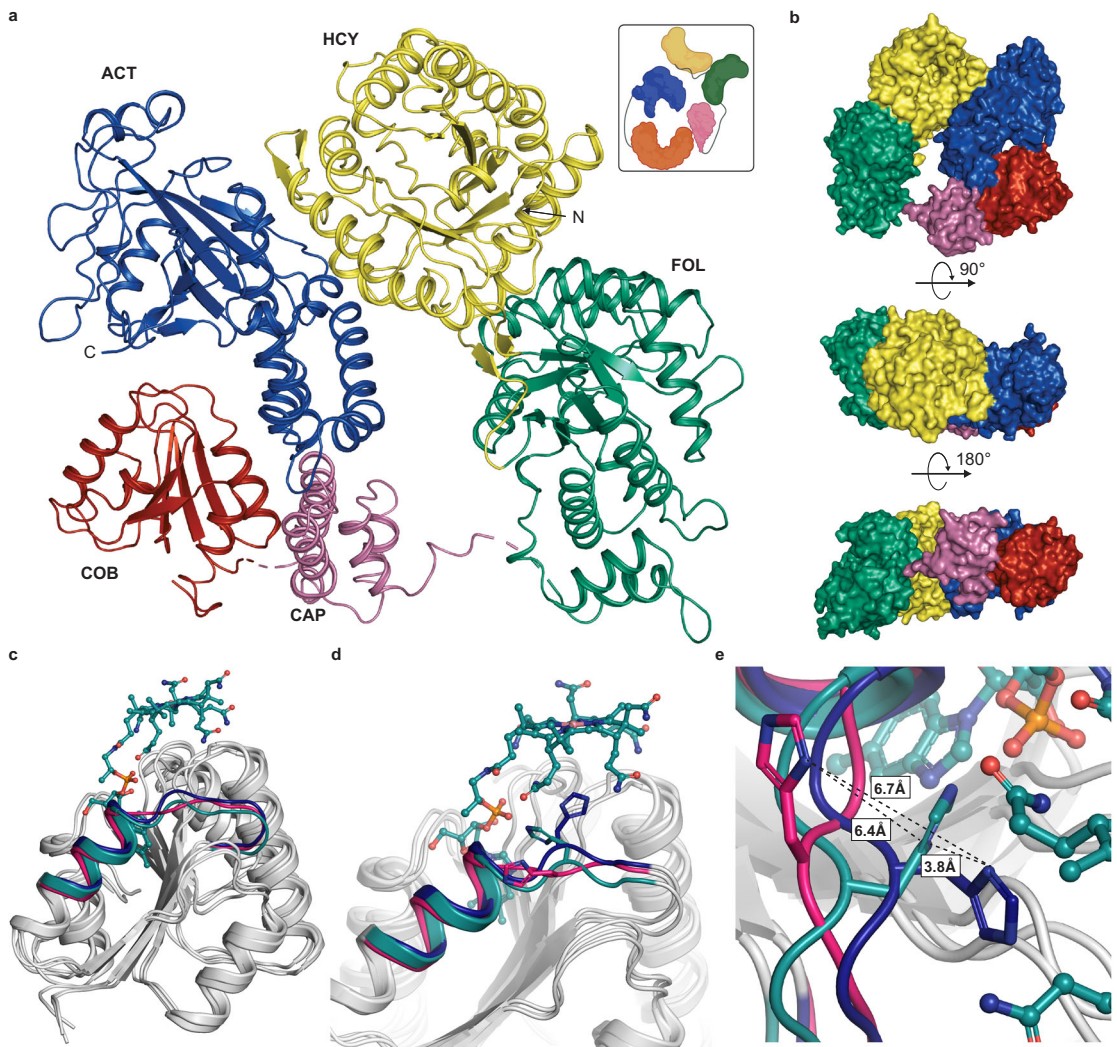

**Fig. 4 | Full-length methionine synthase domain organization and local changes upon cofactor loading. a** Structure of full-length cobalamin-dependent methionine synthase from *Thermus thermophilus*. Homocysteine binding domain (Hcy) shown in yellow, Folate binding domain (Fol) shown in green, Cap is shown in pink, Cobalamin binding domain (Cob) shown in red, and the Activation domain (Act) in blue. **b** Surface representations of full-length MS with same domain color as (**a**). **c** Superposition of the *apo* Cob domain from *t*MS (8SSC, gray, hot pink helix) with the Cob domains from *Apo*-Cap:Cob:Act (8SSD, gray, blue helix) and *Holo*-

Cap:Cob:Act (8SSE, gray, teal helix) bound to Cbl(II) (teal) showing restructuring of the flexible loop on the C-terminus of helix α1 containing His761 upon cobalamin binding. **d** Alignment of Cob domains with conserved His761 (shown in sticks) in MS with the same colors as (**c**). **e** Side view of the flexible loop on the C-terminus of helix α1 containing His761 (shown in sticks) in MS with the same colors as (**c**). Distances between each respective His761 residue are shown. Panel **a** insets were created with BioRender.com.

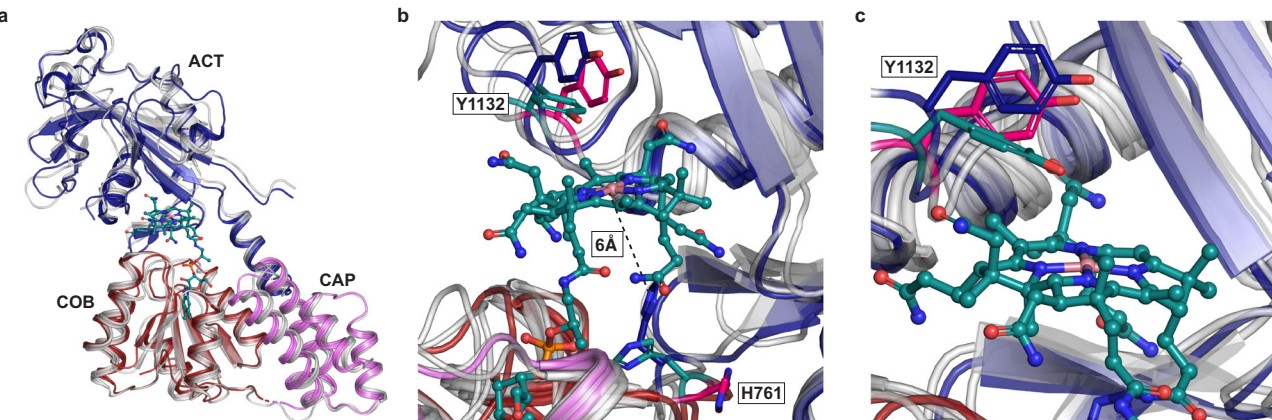

**Fig. 5 | MS captured loading Cobalamin *in crystallo* highlights the flexibility of His761 and Tyr1132. a** We captured the His-off, base-off binding of cyanocobalamin by *t*MS. Only Cap, Cob, and Act domains from the *apo*-full-length are shown in gray, while in the *holo*-Cap:Cob:Act tridomain the Cob domain is shown in red, the Cap domain in pink, and the Act domain in blue. Despite being observed in the Act-on state in its *apo*-form, cobalamin was found to be bound in the binding pocket with minimal global rearrangements observed from its resting position.
**b** Interactions between the cofactor (teal), displaying the flexible nature of His761

between *apo* structures: full-length (hot pink), *apo*-tridomain (blue), and the *holo*-tridomain (teal). Tyr1132 displays its flexibility on the upper-axial portion of the cofactor, rotating to serve as a "lid" in the *holo* structure as compared to the *apo* structures. The distances between the His761 residue of the *apo*-tridomain and the Cbl cofactor present in the *holo*-tridomain are shown. **c** Side view of the flexible nature of Tyr1132 over the upper-axial portion of the cobalamin cofactor shown with the same colors as (**b**).

as previously determined with the structure of the C-terminal MS fragment, that includes the Cap, Cob, and Act domains, trapped in the reactivation conformation (Supplementary Fig. 6)[40–42]. Notably, *t*MS crystallized in the *apo*-form; despite the absence of Cbl, the Cob domain retains the same Rossmann-fold demonstrated by the superposition of *apo*-*t*MS Cob domain with the Cbl-bound Cob domain of *E. coli* MS structures that have so far been determined (Supplementary Figs. 5, 6). The four helical bundle Cap domain is displaced ~25 Å from the resting Cap-on state and virtually identical in form and position to previously captured Cap-off structures (Supplementary Figs. 5, 6). As such, cofactor entry is not occluded and its binding site is solvent accessible, with the Cob domain found to be preformed and ready to bind Cbl requiring minimal rearrangements.

The tandem TIM-barrel Hcy and Fol domains are poised adjacently, separate from the Cob:Act domains and align well with previous structures of either the excised Fol domain or with the N-terminal fragment consisting of the Hcy and Fol domains (Supplementary Fig. 5). The Fol domain is characterized by a unique TIM-barrel[45]. In contrast to the canonical $\beta_8\alpha_8$-barrel, the Fol domain is a $\beta_8\alpha_7$-motif where the last antiparallel helix-turn-helix motif is replaced by a helical bundle that interacts with the exterior of the domain. This fold is highly conserved and essentially identical in the excised Fol domain (5VON) and full-length structure (RMSD 0.46 Å, Supplementary Fig. 5).

The Hcy and Fol domains move together as a rigid tandem body that locks the relative orientation of the substrate binding sites[44], where an interdomain linker (Linker I, Fig. 1c) serves to 'zip' the two domains together, effectively securing them in place[44]. The edge of the Hcy domain rests against the distal side of the Act domain (Fig. 4a), forming extensive interdomain contacts through >25 hydrogen bonds and salt bridges resulting in a buried surface area of ~900 Å. Computational (PISA) analysis of this interface though indicates that these exclusively electrostatic interactions, especially for a thermophilic protein, are consistent with the transient nature of this interface allowing for the conformational rearrangements[58] that are required to support the three methyltransferase reactions. Two out of four linker regions, Lys649-Asp657 (Linker II, Fol:Cap) and Glu877-Ala897 (Linker IV, Cob:Act) could not be modeled due to a lack of sufficient electron density, though the other linker regions, Pro342-Asn352 (Linker I, Hcy:Fol) and Met741-Gly747 (Linker III, Cap:Cob), could be modeled. The side chain electron density of Linkers II and IV indicated that the

linker region was mobile, emblematic of the dynamic nature of these linkers. The flexible linker regions, particularly for Linker II (Fol:Cap) and Linker IV (Cob:Act), indicate that in the reactivation conformation, they could be key in orchestrating domain movements in the MS global conformational ensemble.

As the only structure obtained for any B12-dependent methionine synthase in its *apo*-form, this structure gives a unique insight into the potential domain movements required prior to and once cobalamin is loaded. One characteristic of a large subset of cobalamin-binding enzymes is that they bind Cbl in the base-off/His-on state, where the dimethylbenzimidazole (DBI) tail inserts into the conserved nucleotide binding cleft of the Cob domain and is replaced by His761 in the lower axial position. His761 is part of the highly conserved DxHxxG motif on a flexible loop on the C-terminus of helix $\alpha 1$ (aa747–761) that forms the upper edge of the lower-ligand binding site[59]. Without Cbl bound, His761 is flipped out, serving to open the nucleotide-binding cleft to facilitate DBI insertion (Fig. 4c, d). Upon Cbl binding, His761 moves into position to interact with the cobalt (Fig. 4d, e). However, how cofactor loading occurs in MS is not known. The stability of *apo*-*t*MS and its ability to bind and load cobalamin in solution affords the opportunity to study direct cofactor loading structurally.

## Structural basis of cofactor loading

To interrogate the structural basis of cofactor loading, we sought to crystallize the Cap:Cob:Act tridomain in its *apo*-form, then performed soaking experiments to obtain the *holo*-Cap:Cob:Act complex. The excised tridomain was chosen to provide a simpler, minimal model sufficient to study cofactor loading. Given that MeCbl is photosensitive, we first sought to explore whether the preformed nature of the Cob domain would allow for cofactor loading of Cbl-analogs *in crystallo*. We selected cyanocobalamin (CN-Cbl, Fig. 1a), a stable and synthetic Cbl that does not support catalysis. To that end, *apo*-Cap:Cob:Act *t*MS crystals were soaked with CN-Cbl, yielding a structure of the *holo*-complex, solved to 2.95 Å resolution and one in the *apo*-form obtained prior to soaking, solved to 2.4 Å resolution (Fig. 5, Supplementary Table 2).

The *apo*-Cap:Cob:Act structure aligns well with the respective portion of the *apo*-*t*MS full-length structure, with an RMSD of 2.34 Å. The Act domain is positioned over the Cob domain, indicating that the *apo*-tridomain structure is found in a conformation that has previously

been associated with the enzyme entering the reactivation conformation (Act-on, Cap-off)[40–42]. However, the loop containing His761, relevant to reactivation and cofactor loading, displays some marked differences. In the *holo*-Cap:Cob:Act structure, the Act domain remains positioned over the Cob domain, above the Cbl cofactor (Fig. 5a). Unexpectedly, the cofactor is found to bind in the His-off, base-off state. Structural alignment with the *apo*-Cap:Cob:Act structure shows that His761 imidazole side chain of the *apo*-tridomain is ~6 Å from the docked Cbl cofactor/Co (Fig. 5b), in an orientation facing away from the cofactor. There is minimal electron density in the upper axial position of the cofactor where the cyano group would reside (Supplementary Fig. 7), with a water molecule positioned 4.5 Å away from the cobalt center; this is in accord with previous reactivation conformation structures of MS. The absence of the cyano group indicates that the Co-CN bond is cleaved, likely via photoreduction during data collection[40,60,61]. Tyr1132 occludes the upper axial face of the Cbl cofactor, much like in previous reactivation structures captured in *e*MS and in contrast to our reported *apo*-*t*MS and *apo*-Cap:Cob:Act structures (Fig. 5b, c). This Tyr residue is universally conserved (Supplementary Fig. 8) and has been shown to be important in reactivation. Mutation of Tyr1139 in *e*MS to phenylalanine lowers the reduction potential of the Co(II)/Co(I) couple, which would not support Co(II) one electron reduction required for reactivation[62]. It is worth noting that CN-Cbl was used in an attempt to observe base-on binding in *t*MS, given that the p$K_a$ associated with the dissociation of its DBI tail is 0.1[63]. As such, the observation of Cbl binding in the His-off, base-off mode, without an axial ligand, was surprising and indicates that the captured Cbl is likely in the Co(II) state, and the cofactor was modeled as such.

The captured *apo* structures (*t*MS full-length and Cap:Cob:Act tridomain) and the tridomain *holo*-structure highlight the preformed cobalamin pocket, which is solvent-exposed and thus readily accessible even *in crystallo*. Comparison between the loaded and pre/unloaded cobalamin domains reveals that upon cofactor loading, His761 movement is limited by its interaction with the DBI tail. Indeed, the orientation of His761 has not previously been observed. His761 forms a hydrogen bond with the phosphate tail of the DBI lower ligand, with the cobalamin bound in a His-off, base-off manner (Supplementary Fig. 8). Additionally, Tyr1132 is observed to swing and "cap" the upper axial portion of the cobalamin cofactor, as opposed to the open/away orientation observed in the *apo* structures (Fig. 5b, c). In order for *t*MS to undergo reactivation, His761 and the helix α1 adjacent loop on which it is found would have to reorient itself to bind cobalamin in the His-on, base-off mode, and Tyr1132 would need to swing and "uncap" the upper axial portion of the cobalamin cofactor. The requirement for reorganization of His761 and Tyr1132, in addition to the need for a reductive partner to reduce cob(II)alamin to cob(I)alamin, priming it for reactivation via methylation by SAM, indicates that the captured *holo*-structure is one en route to reactivation. Furthermore, the captured *apo*-Cap:Cob:Act structure would fulfill the reorganization requirements, priming the cobalamin cofactor for reductive methylation.

## Discussion

We have discovered a thermophilic functional methionine synthase homolog that is more suitable and amenable to structural, biochemical, and mechanistic characterization. *t*MS can be purified in the absence of cobalamin, is highly heat-tolerant, allowing for quick and facile purification, and is highly tolerant to mutagenic studies, with all variants constructed so far displaying remarkable stability, all factors that allow for robust biochemical and structural studies. *t*MS can be reconstituted with non-native cobalamins, including but not limited to aqua-Cbl, CN-Cbl, and *n*-propyl-Cbl, either with a simple incubation step or even *in crystallo*, opening the door for the use of cobalamin mimics and analogs that will facilitate mechanistic interrogations.

The ability to study *t*MS in its *apo*-form, devoid of any Cbl cofactor, allowed for biochemical and structural interrogation of

cofactor loading independent of any substrates. Previous work on excised MS domains showed that the substrate binding sites were preformed, accommodating substrates with minimal changes in the microenvironment lining the substrate pockets, amounting to predominantly local versus global changes, as compared to their *apo*-form. The *apo*-full-length and *apo*-Cap:Cob:Act *t*MS structures, when compared to the *holo*-Cap:Cob:Act structure, show that the cobalamin binding pocket is preformed in a similar manner. However, in the case of Cbl binding, His761 was found in a unique orientation, interacting with the ribosyl tail of the Cbl cofactor. This (re)arrangement adds to the list of observed His conformations for Cbl-binding proteins and shows the intricate role His plays in potentially telegraphing the overall Cbl coordination and oxidation state.

The increased stability of *holo*-*t*MS versus its *apo*-form shows that despite minimal structural rearrangements upon cofactor binding, the changes are significant enough to stabilize the structure as a whole (Fig. 2a). How each domain communicates after ligand binding events appears to be subtle, but dependent on universally present linker regions that allow for stochastic but likely prearranged conformational sampling that must pass several checks and balances. Cobalamin must be in the proper oxidation state; MTF must be bound in order to "trigger" a domain rearrangement that would place the Cob domain in proper position to react; HCY, $Zn^{2+}$, and $K^+$ must be bound prior to domain rearrangement(s) required for the Cob domain to interact with the Hcy domain after interacting with the Fol domain. Cobalamin-binding enzymes can show high avidity once Cbl is bound, and Cbl is not released: this prevents highly reactive Cbl species from being subject to inactivation or futile cycling, and instead, the reactive intermediate is shuttled to the next relevant domain for catalysis[64].

Given that MS can adopt numerous conformations in solution, exactly how MS regulates these domain arrangements remains an open question in the field. Truncated MS structures have predicted substantial domain motions of MS in its catalytic and reactivation cycles but have so far revealed only minimal local rearrangements upon substrate binding. We posit that the Cob domain coupled with its cofactor state plays a central role in controlling these conformations for catalytic function. Our full-length MS structure and predicted hypothetical model conformations (Supplementary Fig. 9) indicate that the Cob domain must interact transiently with either the Act domain or the Hcy:Fol domains at different points in the catalytic and reactivation cycles. Moreover, the full-length structure recapitulates the finding that the Hcy:Fol domains are locked in place by Linker I, effectively making the Hcy:Fol domains move as rigid bodies[44]. The reactivation conformations, found in the Act-on and Cap-off state, lead to a Cap:Cob linker (Linker III) that is more rigid and is thus observed in our structures. This is in contrast to the Cap-on Cap:Cob didomain structure which lacks the Act domain (1BMT), where the corresponding linker could not be modeled due to its increased flexibility. The observed flexibility of the other linker regions, particularly for Linker II (Fol:Cap) and Linker IV (Cob:Act) indicate they could be key in orchestrating domain movements in the MS conformational ensemble, guided by the helix α1 adjacent loop containing His761 acting as a molecular switch.

Consequently, we propose that the Cob domain engages the appropriate substrate-binding domain to form a catalytically competent ternary complex and active site, with the Cbl cofactor's oxidation and alkylation status functioning as a gatekeeper. The status of the Cbl cofactor varies for each methyl transfer reaction, MeCbl (Co(III) oxidation state) for homocysteine methylation, Co(I) for accepting a methyl group from MTF, and Co(II) for reductive methylation using SAM as the methyl donor (Fig. 1b, e, Supplementary Fig. 1). The His761 residue of the Cob domain coordinates with the Co atom of Cbl, and any resulting change in the His coordination of Cbl, along with the coordination and rearrangement of the His residue itself can serve as a molecular signal for the Cob domain to report on the status of Cbl

required for catalysis (e.g. differences in the upper ligand and the oxidation state of Co) (Figs. 4d, e and 5b, c). Rather than relying on protein:protein domain interactions, cobalamin appears to be the primary mediator governing which substrate-binding domain can form a ternary complex with the Cob domain, with linker regions and His761 playing a role in guiding the conformation ensemble.

The ability to probe mechanistic questions about how methionine synthase achieves three chemically challenging methylations through structural analysis allows for a more granular approach towards understanding MS function. In addition, this work sheds light on outstanding questions that have been the subject of research for decades, and the insights gleaned here serve to expand our understanding of MS and explore MS as a biocatalytic tool.

## Methods

### *t*MS expression vector construction

A clone of *T. thermophilus* methionine synthase (*t*MS) (NCBI Gene Locus tag TTH_RS03220 [https://www.ncbi.nlm.nih.gov/gene/3168986], GenBank accession code NC_006461.1 [https://www.ncbi.nlm.nih.gov/nuccore/NC_006461.1?from=587649&to=591206]) was obtained from Riken BioResource Center[65]. The gene, originally cloned in pET-11a, was subcloned into pMCSG7 vector using the ligation independent cloning technique (LIC). As a result, *t*MS was expressed as an N-terminal His-tagged form with a TEV site, allowing for tag-removal if desired. The expression vector was designated pMCSG7(*t*MS^wt). To express the truncated *t*MS, pMCSG7(*t*MS^ΔN35) was constructed. *t*MS^Cap:Cob:Act was prepared in a similar fashion, designated pMCSG7(*t*MS^Cap:Cob:Act), encompassing only the Cap, Cob, and Act domains of *t*MS. A complete list of the bacterial strains, plasmids, and primers used in this study are listed in Supplementary Table 4.

### Expression of *t*MS constructs

*BL21star(DE3)* was used for protein expression. *E. coli* transformed with pMCSG7(*t*MS^wt) was propagated at 37 °C in Luria Broth containing 50 μg/mL ampicillin, and protein overexpression was induced using either Isopropyl β-D-1-thiogalactopyranoside (IPTG) (final concentration of 0.1 mM) or using autoinduction media. Cells were grown at 37 °C for 4 h prior to harvesting via centrifugation and stored at −80 °C.

*E. coli* transformed with pMCSG7(*t*MS^ΔN35) or pMCSG7(*t*MS^Cap:Cob:Act) were propagated at 37 °C in Luria Broth containing 50 μg/mL ampicillin, and protein overexpression was induced using autoinduction media. Cells were grown at 30 °C overnight prior to harvesting via centrifugation and stored at −80 °C.

### *t*MS protein purification

For holoenzyme purification, the harvested cell pellet was first resuspended in 50 mM potassium phosphate buffer (KPB) (pH 7.4), 0.3 M sodium chloride, (4–5 mL per 1 g of pellet), to which lysozyme (0.1 mg/mL) and PMSF (1 mM) were added. The resuspended cell pellet was lysed via sonication (4 °C, 5 s on, 5 s off, 5 min total). The crude lysate was centrifuged (15 min × 3, 10,000 × *g*, 4 °C) decanting the supernatant to remove any cellular debris/pellet. Methylcobalamin (MeCbl) (Millipore Sigma) was added to the crude supernatant (20 mM) to form *holo-t*MS. The crude extract with MeCbl was incubated at 70 °C for 15 min, then centrifuged (15 min, 10,000 × *g*, 4 °C). The supernatant was pooled and filtered via vacuum (0.45 μm), to which imidazole was added (20 mM) and the supernatant was applied/loaded onto a Ni-affinity column (His-trap Chelating HP, Cytvia, 5 mL) equilibrated with 50 mM KPB, pH 7.4, and 20 mM imidazole. The column was washed with 50 mM KPB, pH 7.4, 0.3 M sodium chloride, 20 mM Imidazole (1 CV), then 50 mM imidazole (5 CV), and the protein was eluted in bulk with buffer consisting of 50 mM KPB, pH 7.4, 0.3 M sodium chloride, and 80 mM imidazole (10 CV) and 150 mM Imidazole (3 CV). Red-colored fractions were collected and subjected to a TEV digest, with

dialysis at 4 °C overnight (50 mM KPB, pH 7.4). The dialyzed sample was incubated at 70 °C for 15 min to quench the TEV digest, then centrifuged (15 min, 10,000 × *g*, 4 °C). The resulting supernatant was filtered (vacuum, 0.45 μm) and concentrated via centrifugation in 25 mM Tris, pH 7.4, 50 mM KCl, 1 mM TCEP; concentrated, purified *t*MS (-10 mg/mL) was stored at 4 °C. The purified enzyme was stable for at least a month, as judged by enzymatic activity (Supplementary Table 3).

Purification of *apo-t*MS^Cap:Cob:Act was conducted similarly, with no added cobalamin. The harvested cell pellet was resuspended in 50 mM KPB (pH 7.4), 300 mM NaCl (-5 mL/1 g of the pellet), to which lysozyme (0.1 mg/mL), PMSF (0.1 mM), and TCEP (1 mM) were added. The resuspended cell pellet was lysed via sonication (4 °C, 5 s on, 5 s off, 5 min total). The crude lysate was centrifuged (15 min × 3, 10,000 × *g*, 4 °C) decanting the supernatant to remove any cellular debris/pellet. The crude supernatant was filtered via vacuum (0.45 μm) and pooled before FPLC purification using a modified IMAC bulk elution (Buffer A: 50 mM KPB, pH 7.4, 300 mM NaCl, 20 mM Imidazole, 1 mM TCEP, Buffer B: 50 mM KPB, pH 7.4, 300 mM NaCl, 250 mM Imidazole, 1 mM TCEP), with the desired protein eluting at 30–100% Buffer B. Fractions containing Cap:Cob:Act as confirmed by SDS–PAGE were then pooled, and an ammonium sulfate cut was used to remove any impurities, using an equal volume of saturated ammonium sulfate (-4 M) to achieve a final concentration of 50% (w/v). The mixture was centrifuged (10,000 × *g*, 10 min, 4 °C) to pellet any precipitated protein. The supernatant was carefully decanted to avoid disturbing the pellet; the pellet was resuspended in 25 mM Tris, pH 7.4, 50 mM KCl, and 1 mM TCEP. A TEV digest (120 rpm, 4 °C, 18 h) followed and was quenched by heating at 60 °C (20 min). Following clarification of the supernatant by centrifugation (10,000 × *g*, 10 min, 4 °C) and filtration (vacuum, 0.45 μm), the clarified supernatant was loaded onto a pre-equilibrated HiLoad 16/600 Superdex 200 pg column (Cytvia) (2 CV of SEC Buffer: 25 mM Tris, pH 7.4, 50 mM KCl, 1 mM TCEP), yielding the desired protein (-50 kDa). The protein was found to elute as a monomer, with minimal to no aggregation visible. Fractions containing the desired protein, as judged by SDS–PAGE, were pooled and concentrated via centrifugation in 25 mM Tris, pH 7.4, 50 mM KCl, 1 mM TCEP to yield purified *t*MS^Cap:Cob:Act (-15 mg/mL), which was stored at 4 °C or flash-frozen for long-term storage at −80 °C.

### *t*MS thermostability analysis

Thermostability and protein stability were examined in the presence and absence of cobalamin. Crude extracts from *E. coli* cells containing *apo-t*MS^wt were incubated at varying temperatures with or without 100 μM CH₃-cobalamin (20–70 °C) for 15 min in 50 mM KPB, pH 7.4, then centrifuged at 20,000 × *g* for 5 min at 4 °C. The amount of protein in the soluble fraction was assessed by analyzing the supernatant using SDS–PAGE followed by staining with Coomassie Brilliant Blue.

### Temperature dependence of *t*MS activity

Non-radioactive assays were employed to measure *t*MS activity with minor modifications[66]. Briefly, the reaction mixture (800 μL total volume) contained 0.5 M DTT, 0.5 mM aquacobalamin (Millipore Sigma), 0.1 M homocysteine, 0.25 mM (6*S*)-CH₃-H₄folate (MTF) (Merck Eprova AG), and 3.8 mM AdoMet (SAM) (Millipore Sigma) in 100 mM KPB (pH 7.2) using 20 nM of protein (*holo-t*MS). Reactions were incubated with the reaction mixture in the absence of folate at 50 °C for 5 min, then initiated by adding MTF. The reaction mixture was quenched after 30 min by adding formic acid–HCl (200 μL), followed by heating at 80 °C for 10 min. Once cool, the reaction formed methenyltetrahydrofolate, and the concentration was determined using the extinction coefficient of $26.5 \times 10^3 \, M^{-1} \, cm^{-1}$ at 350 nm. The assays were conducted at 25–70 °C for 0–30 min to determine time and temperature-dependent activity.

### Substrate affinity and dependence

Assays to assess substrate affinity and dependence were conducted at varying substrate concentrations and run in a similar fashion. Assays were performed with 63 nM *holo-t*MS$^{wt}$ where the concentration of (6*S*)-methyltetrahydrofolate was varied from 0 to 104 μM, or where the concentration of DL-homocysteine varied from 0 to 57.5 μM. These reactions were incubated at 50 °C for 5 min, quenched by adding formic acid−HCl (200 μL), followed by heating at 80 °C for 10 min.

Assays to assess *t*MS turnover and SAM dependence were conducted with 25 nM *holo-t*MS$^{wt}$ using 1.25 mM DL-homocysteine, 104 μM (6*S*)-methyltetrahydrofolate, and 0–10 μM of *S*-adenosylmethionine, then incubated at 50 °C for 0–20 min. Reactions were quenched by adding formic-acid/HCl mixture (200 μL) followed by heating at 80 °C for 10 min.

All the measurements were conducted in triplicate, and the error values are indicated by standard errors.

### Crystallization of *t*MS

Crystals were grown via sitting drop vapor diffusion. Purified *apo-t*MS$^{ΔN35}$ (~15 mg/mL in 25 mM Tris, pH 7.5, 50 mM KCl, 1 mM TCEP) was mixed with 0.2 M trimethylamine *N*-oxide (TMAO), 0.1 M Tris−HCl (pH 8.5), 20% (w/v) polyethylene glycol monomethyl ether (PEG-MME) 2000 in a 1:1 ratio (1 μL each) and incubated at 4 °C. Crystals were briefly transferred to a cryo-protectant solution containing glycerol (20%) and 0.16 M trimethylamine *N*-oxide (TMAO), 0.08 M Tris−HCl (pH 8.5), 16% (w/v) polyethylene glycol monomethyl ether (PEG-MME) 2000 prior to harvesting and flash freezing in liquid nitrogen.

Purified *apo-t*MS$^{Cap:Cob:Act}$ (~15 mg/mL in 25 mM Tris, pH 7.5, 50 mM KCl, 1 mM TCEP) was mixed with 0.1 M sodium acetate (pH 4.6) and 2 M sodium formate in a 1:1 ratio (1 μL each) and incubated at 4 °C. Crystals were briefly transferred to a cryo-protectant solution containing ethylene glycol (10%), 0.09 M sodium acetate (pH 4.6), and 1.8 M sodium formate prior to harvesting and flash freezing in liquid nitrogen.

Purified *apo-t*MS$^{Cap:Cob:Act}$ (~15 mg/mL in 25 mM Tris, pH 7.5, 50 mM KCl, 1 mM TCEP) was mixed with 0.1 M Bis−Tris Propane (pH 7), 60% (v/v) Tacsimate (pH 7), and 5% PEG 300 in a 1:1 ratio (1 μL each) and incubated at 4 °C. Non-soaked crystals were harvested without additional cryoprotection then flash frozen in liquid nitrogen. *Apo-t*MS$^{Cap:Cob:Act}$ crystals to be soaked with cyanocobalamin were first transferred to a solution consisting of 0.1 M Bis−Tris Propane, pH 7, 60% (v/v) Tacsimate, pH 7, 5% PEG 300, 10% ethylene glycol, ~1.5 mM cyanocobalamin (Millipore Sigma), in which they were stored for 1–5 min prior to being flash-frozen in liquid nitrogen.

Data collection and processing statistics are summarized in Supplementary Table 2. Data for *apo-t*MS$^{ΔN35}$ were indexed to space group $P4_12_12$ (unit-cell parameters $a = b = 134.84$, $c = 174.74$ Å) with one molecule in the asymmetric unit (Matthew's coefficient VM = 3.12 Å$^3$ Da$^{-1}$, 60% solvent content). Data for *holo-t*MS$^{Cap:Cob:Act}$ were indexed to space group *C*121 (unit-cell parameters $a = 166.1$, $b = 95.8$, $c = 238.7$ Å) with six molecules in the asymmetric unit (Matthew's coefficient VM = 2.70 Å$^3$ Da$^{-1}$, 54% solvent content). Data for *apo-t*MS$^{Cap:Cob:Act}$ were indexed to space group $P3_12_1$ (unit-cell parameters $a = 96.18$, $b = 96.18$, $c = 356.04$ Å) with three molecules in the asymmetric unit (Matthew's coefficient VM = 2.71 Å$^3$ Da$^{-1}$, 55% solvent content).

### Data collection and refinement

X-ray data sets were collected at 100 K on GM/CA beamline 23-ID-B at the Advanced Photon Source, Argonne National Laboratory (Argonne, IL). Data sets were processed using xia2/DIALS[67]. Initial phases for *t*MS$^{ΔN35}$ were obtained using Phaser[68]. For search models, the Hcy and Fol domains (PDB 1Q7M, 5VON) without substrates were used as N-terminal rigid bodies while the *C*-terminal domains were separately treated as rigid bodies, using the Cob domain (PDB 1BMT) with no

cofactor, and the Act domain (PDB 1MSK) with no substrate for the molecular replacement method. The Hcy and Fol domains were placed first, followed by the Act domain, then the Cob domain. Iterative model building and corrections were performed manually using Coot[69] following molecular replacement, with the Cap domain being placed manually, and subsequent structure refinement was performed with CCP4 Refmac5[70]. PDB-REDO[71] was used to assess the model quality in between refinements and to fix any rotamer and density fit outliers automatically. The model quality was evaluated using MolProbity[72].

For *t*MS$^{Cap:Cob:Act}$, the Cap:Cob:Act domains from *t*MS$^{ΔN35}$ were used as the search model for molecular replacement using Phaser[68]. Data sets were processed using xia2/DIALS[67]. Iterative model building and corrections were performed manually using Coot[69] following molecular replacement and subsequent structure refinement was performed with CCP4 Refmac5[70]. Initial refinement was conducted using BUSTER[73] to rapidly fix Ramachandran, rotamer, and density fit outliers, refining to convergence and adding waters in the final automated round of refinement. For *holo*-Cap:Cob:Act, Phenix eLBOW[74] was used to generate the initial ligand restraints using ligand ID "B12". Phenix LigandFit[75] was used to provide initial fits and placements of the B$_{12}$ ligands. PDB-REDO[71] was used to assess the model quality in between refinements, to fix any rotamer and density fit outliers, and to optimize the ligand geometry and density fit automatically. The B$_{12}$ ligand restraints were edited manually to minimize ligand geometry outliers; the best fit with minimal geometry outliers according to the PDB validation server was obtained using B$_{12}$ ligand restraints provided by BUSTER. The model quality was evaluated using MolProbity[72]. The *apo*-Cap:Cob:Act structure data set showed partial crystal twinning and final rounds of refinement were conducted using twin refinement, as suggested by PDB-REDO. Figures showing crystal structures were generated in PyMOL[76].

### Statistical analysis and reproducibility

Unless otherwise stated, functional assays were conducted using $n = 3$ independent replicates. At least three independent experiments were conducted for each functional assay. Data were analyzed as mean ± SEM. Analysis and curve-fitting was performed using Prism 9.5.1.

### Reporting summary

Further information on research design is available in the Nature Portfolio Reporting Summary linked to this article.

## Data availability

The structure coordinates and structure factors reported in this study have been deposited in the Protein Data Bank under accession codes 8SSC (*t*MS$^{ΔN35}$), 8SSD (*apo-t*MS$^{Cap:Cob:Act}$), and 8SSE (*holo-t*MS$^{Cap:Cob:Act}$). PDB codes of previously published structures used in this study are 1Q7M, 1K7Y, 1K98, 1BMT, 1MSK, 2O2K, 3IVA, 3IV9, 3BUL, 3BOL, 3BOF, 4CCZ, 5VOO, 5VOP, 5VON, 6BM5, and 8G3H. All other data are available from the corresponding authors upon request. Source data are provided with this paper.

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

## Acknowledgements

The authors acknowledge GM/CA and LS-CAT beamlines at the Advanced Photon Source for beamtime. Figure cartoons (Figs. 1c, d, 4a inset, and Supplementary Fig. 9, insets) were created with BioRender.com. This work was funded by Rackham Merit Fellowship (J.M.) and the National Science Foundation (CAREER 194517 to M.K.).

## Author contributions

J.M., M.P., K.Y., and M.K. contributed to writing the paper. J.M. and K.Y. expressed and purified proteins and performed biochemical analysis. J.M., M.P., and K.Y. contributed to the protein crystallization. J.M., M.P., K.Y., and M.K. collected crystallographic data. J.M., M.P., and M.K. solved the structures, and J.M., M.P., K.Y., and M.K. performed structural analysis. All authors edited and approved the final version of the manuscript.

## Competing interests

The authors declare no competing interests.
