## [Peer Review File · Nature Communications]

Reviewers' Comments:

Reviewer #1:

Remarks to the Author:

The cobalamin dependent methionine synthase (MetH) is a complex multimodular protein that catalyzes the key step in the biosynthesis of methionine. The cobalamin cofactor is employed as a way-station for the methyl group that is derived from methyl tetrahydrofolate (MeTHF), on its way to Hcy to form Met. The cofactor cycles between the +1 and +3 oxidation states, as it undergoes methylation by MeTHF and demethylation by Hcy, respectively. The cofactor is periodically oxidized to the catalytically inactive +2 oxidation state. To return the protein to the active form, MetH from many sources also have an additional domain that binds S-adenosyl-L-methionine (SAM), which is used to reductively methylate the cofactor. MetH serves an important role in mammalian physiology. Mutations that impair with various aspects of MetH activity are associated with various disease states.

The domains that bind various players in the activity of MetH have long been thought to be arranged as "beads on a string" with hinge regions that connect each of them. Pioneering studies by the labs of Rowena Matthews and the late Martha Ludwig established many of the basic structural and functional outlines of catalysis by this enzyme. Notably, individual domains were shown to be able to function independently, and when combined, recapitulate the overall activity of the protein, in trans. Structural studies on these various domains, starting with the first structure of a B12 binding domain solved in the early 1990's, have established the structural features of each of the active sites. However, the mechanism by which these domains interact with one another and the rules that dictate how each domain interacts with the cobalamin cofactor, in turn, have remained elusive.

This paper by Koutmos, Yamada, and coworkers established some of the key structural details that have been missing in the MetH field. Notably, the authors show the X-ray crystal structure of the full apo-MetH, for the first time showing how the domains are arranged relative to one another. Notably, the SAM-binding domain is situated above the B12-binding Rossmann fold, ready to accept and methylate B12. Using a smaller version of the protein they also show that the protein can be reconstituted with B12, and provide a snapshot of this initially reconstructed structure. These are very important insights.

MetH plays an important role in physiology and so even after decades of work on the protein, much remains to be understood, and 2023 is likely to be considered a watershed year in terms of structural details of MetH. This is the second paper this year on this topic, the first being a cryoEM/SAXS study from the Ando lab, which appeared in PNAS last month. Both studies employ a thermophilic homolog of the protein. While the overall impact of this work is somewhat dulled by the Ando work, this paper has several key strengths. First, the Ando study of the full length protein was at a lower resolution, and the SAM-binding domain was not well positioned in the structure. Second, the Ando study did not provide any insights into the cofactor delivery. Finally, the biochemical tools to express and purify MetH and its various domains that are established in this paper are likely to be transformative in additional studies. Therefore, the impact of the present study is significant.

Overall, the study is straightforward and presented in a logical manner. However, there are several areas where I think the manuscript can be improved. In general, I would recommend a careful reading of the entire manuscript and checking for consistency. I felt that different sections were likely written by different folks and there were some consistency issues.

I apologize for not including page or line numbers. The printout on which I made all my changes cut them off. I am including enough context for each comment so that they can be located readily.

- Abstract, line 2 – "...and S-adenosyl-L-methionine"
- Introduction, line 3 – cobalt
- Introduction, 2nd paragraph – "...Co(I) state of the cofactor..."

- Introduction and throughout the manuscript – please be consistent with the abbreviations. Use MeTHF for methyltetrahydrofolate, Met for methionine, Hcy for homocysteine, etc. Also take a look at the figures and make sure that the abbreviations for the pieces of the protein (Cap, etc.) are consistent throughout.
- The following statement in the introduction is too cryptic and will be confusing to the reader: "...controlled by association and dissociation of different axial ligands to achieve each reaction...". It will be helpful here to expand this statement a little bit to catch all the nuances that are buried.
- The comment in the introduction "...along with the dearth of biochemical data, has raised further...". I think this is misleading. Yes, there isn't a log known about the cofactor loading, but quite a bit is known about chaperones, about the domains and their boundaries, etc. that this comment really does disservice to a lot of great biochemistry that has come out of Michigan in on this topic.
- "including homologs from...Escherichia coli..."
- "...including around the catalytic His761. That I s found...". Please expand this a little bit and provide some context. This residue modulates reactivity of the cobalamin and is not strictly a catalytic residue.
- "...HB8 genome encodes a five-module..."
- "Despite relatively high levels of sequence conservation...". What do you mean about relatively high? Be quantitative.
- The sections dealing with the biochemical data in Figure 2 are very confusing. Are these assays looking at reactions in trans, the way Goulding et al did them? What is really going on in the reactions? Why does the assay in Fig 2d plateau – is it because it is running out of substrate(s) or because the enzyme is getting inactivated? Are the various forms shown in Fig. 2c active in the expected reactions? Are they active in trans?
- "...wreath-like form, and each domain...". This statement needs some references.
- Figure 4 legend – Histidine 761 should be abbreviated His761.
- ...all factors that allow for robust biochemical and structural studies. tMS can be reconstituted with non-native cobalamins...". Please be specific here. I think that you are referring to the CN-Cbl reconstituted protein here as an example of non-native cobalamin. However, as the authors are aware there are a lot of other types of cobalamins that could have been placed here and I would bet that certain ones will not work. Perhaps spell out what you mean so as not to make a broad claim that is not supported by the data presented.
- "...interacting with the ribotyl tail...". Technically, it is ribityl if the ribose was actually opened, as it is in flavin. In this case, it is the "ribosyl tail..."
- In the Materials, check for consistency. I saw E. coli not italicized. The methyl tetrahydrofolate is already defined earlier – use the abbreviation. In the heading "Thermus thermophilus MS" tMS is already defined – use it.
- Supporting Figure 3 – Which PDB structures? Provide PDB accession numbers and references
- Supporting Figure 4 – This figure is completely unnecessary and inappropriate. This manuscript does not do anything with the human or E. coli proteins. So why is half of the figure showing those? In the second half, there are Xs where there should be numbers. With and without cofactor should be above the arrow not a distinct state. Etc. "Obtain holo-MS with cobalamin bound" is redundant – holoMS is the cobalamin bound version. I recommend omitting this figure altogether.
- Supplementary Figure 5 – modify the title to "Acid-catalyzed conversion of tetrahydrofolate to methenyltetrahydrofolate". Should probably include a reference to the procedure here and in the

methods.

- Supplementary Figure 6 – “Structural alignment of excised...”
- Supplementary Table 1 – Is the reference format correct?
- General comment on the PDB validation files – I am not a structural biologist but the validation files are clearly showing parameters that are in the “red zones” because of significant numbers of outliers. I appreciate the fact that MS is a large protein and refinement is probably a real pain. But, I think that the authors should either comment on these, or make an attempt to improve the statistics either by additional round of refinement, or perhaps processing at a lower resolution.

Reviewer #2:

Remarks to the Author:

This manuscript reports the isolation, characterization, and structure of cobalamin-dependent methionine synthase (MS) from *Thermus thermophilus*. MS performs three distinct methylation reactions on the substrates homocysteine, folate, and S-adenosylmethionine, and deficiencies in the human enzyme lead to pathologies and birth defects. Beyond its reactions, MS also undergoes periodic reactivation in which an inactive Co(II) species is reduced and methylated to reform the active cofactor. MS is a complex enzyme with five different domains, and the full length protein has been difficult to work with. By using the *T. thermophilus* homolog, the authors were able to crystallize and determine the structure of full-length MS in the apo form (without cobalamin), representing the first full-length structure and the first without cofactor. In addition, the structure of a three domain construct soaked with cyanocobalamin was determined. The structures reveal the overall arrangement of the domains, their interfaces with one another, and varying positions of key residue His791 and Tyr1132. The positions of these residues in the soaked structure suggest that the structure represents an intermediate on the way to reactivation. The work represents a step forward for the field and the *T. thermophilus* system is promising for future studies.

Specific comments:

lines 57 and 65: the Cob domain appears to be orange, not red

lines 89-90: it would be helpful to add a sentence summarizing what excised domain structures are available besides the study in ref. 38. This information is in SI Fig. 6, but would help here to introduce the state of the field.

line 204: please explain what the distances in panel 4e are indicating.

lines 231-233: His761 appears to be on a loop and not on helix $\alpha 1$, which appears to maintain the same position in all the structures. What evidence indicates that the helix is more flexible when His761 is flipped out? Are the B-factors different? Are less residues involved in helical hydrogen bonds?

line 261: " Compared to the apo-Cap:Cob:Act structure, the His761 imidazole side chain is $\sim 6 \text{ \AA}$ from Co" - this comparison is not clear as there is no Co in the apo structure

Why was the structure of the holo form not obtained without soaking? How do the authors know that the soaked structure accurately reflects what happens in solution?

ref. 38 is missing information

Aug. 10th, 2023

Structure of the first full-length cobalamin-dependent methionine synthase and cobalamin cofactor loading captured *in crystallo*

Manuscript Number: NCOMMS-23-25603

Response to reviewers

(Referee comments in black; responses in red)

Referee 1

1. The following statement in the introduction is too cryptic and will be confusing to the reader: “...controlled by association and dissociation of different axial ligands to achieve each reaction...”. It will be helpful here to expand this statement a little bit to catch all the nuances that are buried.

We appreciate the reviewer’s comments and therefore we have expanded the discussion of the role of the axial ligands of cobalamin to include specific examples for both the upper and lower axial ligands observed in MS.

2. The comment in the introduction “...along with the dearth of biochemical data, has raised further...”. I think this is misleading. Yes, there isn’t a log known about the cofactor loading, but quite a bit is known about chaperones, about the domains and their boundaries, etc. that this comment really does disservice to a lot of great biochemistry that has come out of Michigan in on this topic.

We have edited this sentence to emphasize the pioneering studies conducted in the Ludwig and Matthews’ labs, including the addition of a more detailed background of the structural data obtained from said studies.

3. “...including around the catalytic His761. That is found...”. Please expand this a little bit and provide some context. This residue modulates reactivity of the cobalamin and is not strictly a catalytic residue.

We agree with the reviewer's comment and added that His761 can also serve as the lower-ligand in MS. We have however changed the text to provide succinct context with additional citations that discuss the multiple roles His761 plays. However, its potential role in acting as a key signaling residue is expanded on in greater detail in the discussion.

4. "Despite relatively high levels of sequence conservation...". What do you mean about relatively high? Be quantitative.

Specific values for the sequence identity of *t*MS relative to *h*MS and *e*MS have been added (33% and 34% identity respectively).

5. The sections dealing with the biochemical data in Figure 2 are very confusing. Are these assays looking at reactions *in trans*, the way Goulding et al did them? What is really going on in the reactions? Why does the assay in Fig. 2d plateau – is it because it is running out of substrate(s) or because the enzyme is getting inactivated? Are the various forms shown in Fig. 2c active in the expected reactions? Are they active *in trans*?

An explicit mention that the full-length enzyme was used for all biochemical data shown has been included, and the Figure 2 legend has been revised to explicitly mention that the full-length enzyme was used in these assays.

The temperature-dependent assay in Fig. 2d does not show a plateau at the three lower temperatures (25 °C, 37 °C, and 50 °C), given that *t*MS displays optimal activity at 70 °C; at this temperature, a plateau is observed, likely due to substrate consumption. If oxidative inactivation was the root cause, a plateau would be expected for all temperatures tested, but this is not observed.

While we appreciate the reviewer's question regarding the truncated domains shown in Fig. 2c, and whether they are active *in trans*, this is beyond the scope of this paper. Though the excised domains are indeed active *in trans*, the main intention in Fig. 2c is to show that *t*MS provides a

robust biochemical model in which all excised domain constructs could be successfully expressed and purified. The figure legend has been edited to emphasize this point.

6. ...all factors that allow for robust biochemical and structural studies. *tMS* can be reconstituted with non-native cobalamins...”. Please be specific here. I think that you are referring to the CN-Cbl reconstituted protein here as an example of non-native cobalamin. However, as the authors are aware there are a lot of other types of cobalamins that could have been placed here and I would bet that certain ones will not work. Perhaps spell out what you mean so as not to make a broad claim that is not supported by the data presented.

While the reviewer makes a valid point on the wording of the sentence, and explicitly mentioning CN-Cbl only would be correct in the context of the paper, we were intentionally vague as to what non-native cobalamins could be loaded given that unpublished data has shown that *tMS* can bind other non-native cobalamins that have not previously been reported. As the reviewer noted, previous work has shown that *eMS* could bind *n*-propyl, thiocyanato-, azido-, chloro-, bromo-, and even CN-Cbl formed *in situ* (Hoover, D. M. *et al.* Interaction of *Escherichia coli* Cobalamin-Dependent Methionine Synthase and Its Physiological Partner Flavodoxin: Binding of Flavodoxin Leads to Axial Ligand Dissociation from the Cobalamin Cofactor. *Biochemistry* **36**, 127–138 (1997)). However, providing an extensive list of cobalamins that were found to bind to *tMS* is beyond the scope of this paper, though previously reported non-native cobalamins were added to narrow our initial statement.

7. General comment on the PDB validation files – I am not a structural biologist but the validation files are clearly showing parameters that are in the “red zones” because of significant numbers of outliers. I appreciate the fact that MS is a large protein and refinement is probably a real pain. But, I think that the authors should either comment on these, or make an attempt to improve the statistics either by additional round of refinement, or perhaps processing at a lower resolution.

We appreciate the reviewer’s comments regarding the PDB validation files. The RSRZ outliers observed in the *apo*-Cap:Cob:Act structure are due to partial crystal twinning, the lower resolution data, and side-chain outliers: the residues and their sidechains are present at or near

flexible linker regions, and we have chosen to include them. The Methods section has been amended to reflect these choices in greater detail.

A section in the methods addressing the refinement and modelling has also been added to address these concerns, namely to mention that the RSRZ outliers observed in the *holo*-Cap:Cob:Act structure are predominantly due to the B12 cofactor and the known discrepancies regarding the PDB refinement constraints for cobalamin (Knox, H.L., Chen, P.Y.T., Blaszczyk, A.J. *et al.* Structural basis for non-radical catalysis by TsrM, a radical SAM methylase. *Nat Chem Biol* **17**, 485–491 (2021)). The loading of CN-Cbl *in crystallo*, followed by the observation that the axial CN ligand was cleaved all suggest that the cobalamin cofactor is found in a heterogenous state, likely in partial occupancy; though we modelled the cofactor as Cob(II) due to the absence of the CN ligand, the PDB ligand restraints were insufficient in reducing RSRZ outliers associated with the ligand, regardless of whether COB or B12 was used. The best fit was obtained using manually edited B12 ligand restraints after being minimized in Phenix. During deposition, when an issue with several RSRZ outliers was identified, we re-refined and redeposited these final models that represent the best models we could obtain after exhausting different refinement strategies.

It should be noted that the structure of the *holo*-tetradomain (8G3H, Watkins, M., Wang, H., Burnim, A. A., Ando, N. Conformational switching and flexibility in cobalamin-dependent methionine synthase studied by small-angle X-ray scattering and cryoelectron microscopy. *PNAS* **120**, (2023)) displays even more outliers in the aforementioned flexible linker regions and those immediately adjacent to them, along with outliers associated the B₁₂ ligand itself.

8. Supporting Figure 4 – This figure is completely unnecessary and inappropriate. This manuscript does not do anything with the human or *E. coli* proteins. So why is half of the figure showing those? In the second half, there are Xs where there should be numbers. With and without cofactor should be above the arrow not a distinct state. Etc. “Obtain *holo*-MS with cobalamin bound” is redundant – *holo*MS is the cobalamin bound version. I recommend omitting this figure altogether.

The reviewer’s comments and rationale for removing this figure are valid. The figure has been removed from the Supplementary Information.

Minor textual suggestions:

The suggested textual changes regarding abbreviations and italics have been made to the manuscript.

Supporting Figure 3 – Which PDB structures? Provide PDB accession numbers and references.

We have provided PDB accession numbers and references.

Referee 2

1. lines 89-90: it would be helpful to add a sentence summarizing what excised domain structures are available besides the study in ref. 38. This information is in SI Fig. 6, but would help here to introduce the state of the field.

The sentence has been expanded to include a summary of the available excised domain structures and their PDB codes as suggested by the reviewer.

2. line 204: please explain what the distances in panel 4e are indicating.

The figure legend for Fig. 4e has been edited to include an explicit mention that the distances shown are of the His761 residues between each of the three structures, and to highlight the flexible nature of His761 as captured in these structures.

3. lines 231-233: His761 appears to be on a loop and not on helix $\alpha 1$, which appears to maintain the same position in all the structures. What evidence indicates that the helix is more flexible when His761 is flipped out? Are the B-factors different? Are less residues involved in helical hydrogen bonds?

We appreciate the reviewer's comment regarding this statement. The mention of helix $\alpha 1$ has been edited for clarity. This was an erroneous comment leftover from a previous draft referencing work by Tollinger *et. al.* (Tollinger, M. *et al.* The B₁₂-Binding Subunit of Glutamate Mutase from *Clostridium tetanomorphum* Traps the Nucleotide Moiety of Coenzyme B₁₂. *JMB* **309**, 777–791 (2001)). Given that the structures were all obtained at varying

resolutions, B-factors associated with the loop containing His761 have not been mentioned and while relative B-factors have been calculated, the sentences discussing His761 and the loop it is present on have been edited to reflect the main point that His761 itself samples several orientations.

4. line 261: " Compared to the apo-Cap:Cob:Act structure, the His761 imidazole side chain is ~6 Å from Co" - this comparison is not clear as there is no Co in the apo structure

We appreciate the reviewer pointing out this source of confusion. The Fig. 5 legend has been edited to include that the distance obtained for the His761 residue of the *apo*-tridomain from the Co center of the cofactor was measured after alignment with the *holo*-tridomain structure. A sentence was added in the text to further clarify this point.

5. Why was the structure of the holo form not obtained without soaking? How do the authors know that the soaked structure accurately reflects what happens in solution?

Holo-full-length and truncated structures (di, tri, and tetradomain containing the cobalamin domain) have been obtained (unpublished results) without soaking; however, in our hands, the *holo*-full-length crystals did not diffract well. The rationale behind soaking was to determine whether cobalamin could bind *in crystallo* and provide a more granular insight into the structural changes that might arise strictly from cobalamin loading. The proof of concept would allow for future studies in which precious non-native cobalamin analogs could be loaded *in crystallo* instead of in solution, where forming the holoenzyme in solution for structural studies would be unfeasible if the holoenzyme could not be crystallized or if obtained crystals did not diffract.

We also conducted qualitative but not quantitative in solution studies that confirm cobalamin loading in the apoenzyme. These preliminary data indicate that CN-Cbl could be loaded, with no indication that the cofactor gets decyanated. The detailed characterization of cobalamin loading in solution is underway and will be included in future work. However, we felt that it

was beyond the scope of this paper, as the main intent was to show that cobalamin loading could occur both in solution and unexpectedly/surprisingly *in crystallo*.

The holoenzyme structure shows Cob(II)alamin bound in a His-off state; though CN-Cbl was used for soaking, the CN ligand was not observed in our structure. The photoreduction of Cbl cofactors upon X-ray irradiation is well established, and in the case of CN-Cbl would form Cob(II)alamin *in situ*, and favor a state where His-off binding would prepare the cofactor for reactivation. UV-Vis spectroscopy of CN-Cbl *holo*-tridomain tMS showed that CN-Cbl was indeed bound, which has been previously established in *eMS* (Hoover, D. M. *et al.* Interaction of *Escherichia coli* Cobalamin-Dependent Methionine Synthase and Its Physiological Partner Flavodoxin: Binding of Flavodoxin Leads to Axial Ligand Dissociation from the Cobalamin Cofactor. *Biochemistry* **36**, 127–138 (1997)).

Minor textual suggestions:

lines 57 and 65: the Cob domain appears to be orange, not red

We used BioRender to create these figures and used vermilion which is a red-orange color. We have updated the text to change the color references to red-orange.

ref. 38 is missing information

We appreciate the reviewer for catching this mistake. The suggested information has been added.

Reviewers' Comments:

Reviewer #1:

Remarks to the Author:

The authors have addressed my concerns.